# Small Pupils Lead to Lower Judgements of a Person’s Characteristics for Exaggerated, but Not for Realistic Pupils

**DOI:** 10.3390/bs12080283

**Published:** 2022-08-12

**Authors:** Wee Kiat Lau, Marian Sauter, Anke Huckauf

**Affiliations:** General Psychology, Institute of Psychology and Pedagogics, Ulm University, Albert-Einstein-Allee 47, 89081 Ulm, Germany

**Keywords:** pupils, person, characteristics, eyes, uncanny valley

## Abstract

Our eyes convey information about a person. The pupils may provide information regarding our emotional states when presented along with different emotional expressions. We examined the effects of pupil size and vergence on inferring other people’s characteristics in neutral expression eyes. Pupil sizes were manipulated by overlaying black disks onto the pupils of the original eye images. The disk area was then changed to create small, medium, and large pupils. Vergence was simulated by shifting the medium-sized disks nasally in one eye. Pupil sizes were exaggerated for Experiment 1 and followed values from the literature for Experiment 2. The first Purkinje image from the eye photos in Experiment 2 was kept to preserve image realism. The characteristics measured were sex, age, attractiveness, trustworthiness, intelligence, valence, and arousal. Participants completed one of two online experiments and rated eight eye pictures with differently sized pupils and with vergence eyes. Both experiments were identical except for the stimuli designs. Results from Experiment 1 revealed rating differences between pupil sizes for all characteristics except sex, age, and arousal. Specifically, eyes with extremely small pupil sizes and artificial vergence received the lowest ratings compared to medium and large pupil sizes. Results from Experiment 2 only indicated weak effects of pupil size and vergence, particularly for intelligence ratings. We conclude that the pupils can influence how characteristics of another person are perceived and may be regarded as important social signals in subconscious social interaction processes. However, the effects may be rather small for neutral expressions.

## 1. Introduction

There is a popular saying that the eyes are the gateway to a person’s soul. Humans are born with a preference of looking at the eyes. Newborns spend a significantly longer time looking at a face with opened eyes than the same face with closed eyes [1]. The preference to look at faces with opened eyes continues to be higher than the preference to look at faces with closed eyes as the child matures [2]. We inevitably look at a person’s eyes during social interaction. Sometimes, we steal quick glances at a stranger’s eyes on the streets. We even make prolonged eye contact with intimate partners and loved ones. Therefore, the eyes are an important aspect of our daily habits.

First impressions are the characteristics we perceive about another person by looking at them [3,4,5]. We also perceive a person’s characteristics by looking at their eyes and eye region, which is the area surrounding the eyes (between the temples and the nose bridge, including the eyebrows). We may judge a person’s sex by looking at their eyes, as females tend to have smaller eyes than males [6]. The eyebrows can quickly change the perceived emotion of the eyes. For example, ˅-shaped brows appear angry and ˄-shaped brows look sad [7]. We perceive a smile as more genuine when we see wrinkles (i.e., Crow’s feet) surrounding the eye’s edges [8]. These wrinkles may also make the eyes look older [9]. Eyebags develop underneath the eyes when a person is sleep deprived. They are often perceived as tiredness [10]. Overall, we perceive some characteristics about a person when looking at their eye region.

Some studies also suggest that we can tell what a person is thinking by looking at where the eyes are gazing. In the Reading the Mind in the Eye’s (RMET) test, characteristics about a person can be perceived by where the eyes are looking [11]. In the experiment, participants picked an adjective to describe what the pairs of black and white eyes were conveying. Participants were generally good at associating what the eyes conveyed when looking at the gaze. Hence, a person’s characteristics are perceived when looking at the eye gaze.

### 1.1. Perceiving Characteristics about a Person from the Pupils and Vergence

The eyes convey much information about a person, besides the gaze alone. Some studies allude to the idea that the characteristics of another person can be perceived by looking at the pupils. These may be achieved by varying either pupil sizes, or pupil positions (i.e., vergence). Varying pupil sizes can influence how the emotional state of a person is perceived. A sad face appeared much sadder for constricted pupils than dilated pupils [12]. Pupil sizes can also influence the intensity of perceived fear [13].

Pupil sizes may also change the perception of abstract characteristics like attractiveness and trustworthiness. For instance, large pupils and large eyes are perceived as more attractive [14,15]. Neutral eyes appear more trustworthy when the pupils are dilated, and less trustworthy when the pupils are constricted [16,17]. These findings are replicated regardless of whether participants saw Caucasian or Asian eyes with neutral expressions.

Eye vergence tells us at which distance a person is fixating. Our eyes rotate nasally inward to focus on a very near object [18,19]. This contributes to the feeling of being looked at which itself suggests close and friendly relationships [20]. Sometimes, we appear cross-eyed if the object is extremely near to us. According to the literature, looking slightly cross-eyed could also influence how we are perceived by others. Individuals with convergence strabismus exhibit symptoms where one or both pupils are constantly turned nasally inward. As a result, they appear less intelligent due to their eye vergence [21].

These studies show that changes to pupil sizes and vergence may influence how specific characteristics are read from the eyes. Some of these studies also show that changes to pupil sizes and vergence, in relation to the full face, influences how we read certain characteristics about a person. It is currently unknown how much this effect can be elicited solely by the eyes and eye region alone. It is also unknown if changes to one aspect of the pupils can influence how several characteristics are read. For instance, do changes in pupil sizes influence perceived attractiveness, trustworthiness, and intelligence?

### 1.2. Study Goal

The goal of the study was to explore the role of the pupils in reading characteristics of a person by looking at the person’s eyes. Previous studies adopted a mixture of full faces and eyes. For our study, we chose to use only the eyes such that other cues from the face would not influence the perceived impressions. Studies that used eye stimuli also tested dynamic pupil changes [20,21]. Since motion captures attention [22], it might be difficult to tell whether participants relied on the motion of dilating/constricting pupils or the pupils themselves to evaluate a person’s characteristics. We circumvent this issue by showing participants only the pupils’ end state (i.e., after dilation/constriction). Thus, we presented the following pupils in the study: big, medium, small, and vergence.

There were two experiments to the study. Both experiments were identical except for the stimuli shown. In each experiment, the pupils were manipulated differently. In Experiment 1, we presented pairs of eyes with black disks substituting the pupils. In doing so, we exaggerated pupil sizes differences, to make sure that pupil size differences can be easily perceived by the observers instantly. Vergence was also exaggerated by shifting medium-sized pupils. In Experiment 2, pupil sizes were varied according to real variations. In addition, the inserted pupils included reflections (i.e., first Purkinje image).

According to the literature, the eyes and eye region can convey information about a person’s sex and age. Differently sized pupils can influence perceived attractiveness and trustworthiness and the emotional state of a person. Vergence can also influence how we perceived intelligence from another individual. Therefore, we measured following perceived characteristics in both experiments: sex, age, attractiveness, trustworthiness, intelligence, realism, familiarity, arousal, and valence. One’s emotional state is commonly measured using arousal and valence as indicators of emotions from two dimensions of the Self-Assessment Manikin (SAM) scale [23]. We added realism and familiarity as additional variables to rule out whether the ratings could be due to memory effects if participants had previously seen the stimuli.

We hypothesized that pupil sizes (large, medium, and small) would influence the ratings for attractiveness, trustworthiness, intelligence, arousal, and valence based on previous literature. Specifically, large pupils should be rated higher in these characteristics than the other pupil sizes, and small pupils would receive the lowest scores [12,17,24]. We expected that vergence pupils (i.e., crossed-eye) would result in low ratings for intelligence than straight looking pairs of eyes [21]. Sex and age served as kind of control, since eye size and wrinkles affect the perceived sex and age of the eyes, respectively [6,9]. Finally, realism ratings should be higher in Experiment 2 using the real variations of pupils with reflections.

## 2. Methods

### 2.1. Participants

All participants were recruited in accordance with the Declaration of Helsinki and the study was approved by the ethics committee of Ulm University within the context of the SPP2199 project (see funding). Respondents provided informed consent prior before taking part in the study. In both experiments, the sampling method was of convenient means (i.e., friends, colleagues, social media). Those from Ulm University were compensated with course credits. Others volunteered for the experiment knowing that there was no monetary incentive at the end of the experiment. A power analysis yields a target sample size, *n* = 71, for our critical hypotheses t-tests, with the standard 0.8 power and alpha of 0.05.

265 participants participated in Experiment 1 between 24 Mar 2021 and 30 May 2021. This sample was composed mainly of young adults (Mean age = 29.59 years old, *SD* = 13.06, range = 18–76). Most were female (*n* = 180, 68.20%), followed by male (*n* = 83, 31.40%), and diverse (*n* = 1, 0.40%). A new group of 110 respondents participated in Experiment 2 between 07 Sep 2021 and 31 Dec 2021. The sample was composed mainly of young adults (Mean age = 23.25 years old, *SD* = 7.04, range = 18–54 years old). Most were female (*n* = 91, 82.70%), followed by male (*n* = 16, 14.5%), and diverse (*n* = 3, 2.70%).

### 2.2. Stimuli

The stimuli were pairs of eyes taken from the FACES database [25]. This database features colored Caucasian faces of young, middle-aged, and old models. Images from the database are set in portrait with the dimensions: 2835 × 3543 pixels, 300 ppi, 8 bits depth. Eight neutral face images (two young females, two old females, two young males, two old males) were selected from the database for the experiments. These faces were used in a previous work, and we reused them for the following experiments [26].

Image manipulation was carried out using GIMP (version 2.10). The eye stimuli for the present study were created by cropping the eyes from the database face images (Figure 1a). Eye region features like eyebrows were excluded. The resulting dimensions for the eyes were 1140 × 226 pixels. The pupils of each eye stimulus were manipulated to create large, medium, small sizes. The general idea was to cover the pupils by overlaying them with disks. Thereafter, the disk’s area was adjusted to form different sizes (big, medium, small). We also created simulated vergence based on medium pupil size to simplify the complexity of the study. The idea was to shift the medium pupil size disks nasally to form the simulated vergence. Shifting the medium pupil size disks did not change the disk’s area. Only the disk’s horizontal position was changed. Thus, we created a total of four (large, medium, small, vergence) images times eight eye models, (32 eye stimuli in total).

In Experiment 1, all models have the same black disks in both eyes (see Figure 1a). The black disks had the following dimensions: large pupils 78 × 74 pixels, medium pupils 46 × 44 pixels, and small pupils 26 × 24 pixels. Simulated vergence was derived by displacing the left medium pupil 16 pixels and the right medium pupil 14 pixels nasally (i.e., horizontally towards the nose). The pupils were not rotated inwards equally since each eye rotates asymmetrically during vergence movements [27,28].

In Experiment 2, the pupil sizes were manipulated independent of the eye model according to previous literature [17]. In particular, the pupil sizes in Harrison, Wilson [12] ranged 60–167% and those in Kret, Fischer [16] ranged 60–140%. We used the range 50–150%. Like in Experiment 1, we first created black disks for the eye. Each eye model has a different pupil size. From here, the disk’s area was then changed based on the following: large pupils +50% area, medium pupils +0% area, small pupils −50% area. The eye makes small rotations when looking at something far [28]. Therefore, we simulated the vergence of looking at far objects by displacing the left medium pupil five pixels nasally. The camera glare was included to preserve the realism of the original photos and to preserve the first Purkinje image (see Figure 1b).

### 2.3. Design

The experimental designs for both experiments were modifications of a previous work [26]. Experiments 1 and 2 were identical except for the stimuli shown. Participants finished only one experiment. In either experiment, participants were tasked to complete the same online survey. The online survey was administered using the EFS Survey by Questback GmbH [29]. The survey format, how it looks, and the questions in each section were the same in both experiments.

There were four survey versions (Figure 2a) since the 32 stimuli were distributed using a Latin-square method. The purpose was to ensure that participants never saw the same eye model again with different pupils, as prior memory could influence responses when seeing the same eye model. There were four sections per survey. The sections were always presented in the same order (Figure 2b). In Section 1, participants reported their arousal and valence. Arousal and valence were measured using the Self-Assessment Manikin (SAM) scale [23]. This section was repeated at the end of the survey. In Section 2, participants rated perceived sex, age, attractiveness, trustworthiness, intelligence, realism, and familiarity. In Section 3, participants rated the perceived emotional state of the stimuli (arousal, valence) using the SAM scale. In Section 4, participants completed questions concerning demographics (sex, age, education).

The eyes were shown in Sections 2 and 3. Each pair of eyes appeared at the top of the screen. The eyes appeared in random order. The eye order was different in Sections 2 and 3. Participants evaluated the eyes and provided their responses by selecting a dial or by sliding a point along the Likert scale (Figure 3). All scales used in the survey were based on the nine-point Likert scale, except when answering the demographics. The Likert scale’s left pole indicated the least (“not at all …”) and right pole indicated the most (“very …”) for the following characteristics: attractiveness, trustworthiness, intelligence, realism, and familiarity. The sex scale was labelled very masculine on the left pole and very feminine on the right pole. The age scale was labeled in 5-year increments, ranging 20–24, 25–29, …, >60 years old. The arousal SAM scale was low arousal on the left, and high arousal on the right. The valence SAM scale was negative valence on the left, and positive valence on the right.

We implemented two attention check questions in Section 1 to ensure that participants understood the SAM scale. Participants must pick the correct adjectives describing high arousal and positive valence. The adjectives for arousal and valence are described later in the manuscript. Participants must match the correct option on the Likert-scale.

The stimulus always appeared at the top of the screen (Figure 3). The question “This pair of eyes seem to me …” was positioned directly below the stimulus, left-aligned to the survey page. The items to be rated were presented center-aligned within the page. Participants who completed the survey on a mobile device were encouraged to use the landscape orientation on their devices for the survey. Participants navigated to the next survey page by clicking a button at the bottom of the page. There was no backward button and participants could not return to previous pages to change their responses. Participants had to answer all of the questions. The message “please answer this question” appeared when participants failed to provide a response.

Participants responded to eight stimuli. Each of the eight stimuli was presented once in Section 2 (sex, age, attractiveness, trustworthiness, intelligence, realism, and familiarity) and once in Section 3 (arousal and valence). The stimuli that appeared in Section 2 and Section 3 were the same, except the stimuli order in each section was randomized. Participants rated the stimuli across 16 pages in the survey. There were three instructions pages in the survey. There were 2 pages for Section 1, and 3 pages for Section 4 (demographics). In total, participants navigated through 24 pages of the survey. The entire survey took approximately 10–20 min.

### 2.4. Procedure

The survey was administered online without extra supervision. Participants took part in the survey by clicking on a weblink. The link was sent through the Ulm University student mailing list and posted on social media (i.e., Facebook and Instagram). The survey was conducted in the German language. Participants saw the welcome instructional screen upon clicking on the survey link. The welcome screen indicated that the survey would last approximate 10–20 min. Participants were told that there were no right or wrong responses for the survey, and that the demographic information collected could not be used to identify them. Informed consent was given by clicking on the “I agree” button to begin the survey.

A new instruction screen was shown. Participants were briefed about the number of pages in the survey and the types of scales they would encounter. Participants navigated to the next survey page by clicking on a button at the bottom of the page. The instruction screen for Section 1 was presented. Participants were introduced to the pictographic SAM scale, along with the adjectives which described arousal and valence. These adjectives were derived from previous literature [30]. The adjectives for low arousal were relaxed, calm, sluggish, clumsy, sleepy, and rested. The adjectives for high arousal were stimulated, excited, turbulent, nervous, awake, and restless. The adjectives for negative valence were worried, annoyed, dissatisfied, moody, sad, desperate, and bored. The adjectives for positive valence were happy, content, satisfied, comfortable, hopeful, and relaxed. Participants completed two attention check questions to ensure that they comprehended the descriptions for arousal and valence scales. Participants were then randomly assigned to one of four survey versions.

The survey started with Section 1 (Figure 2b). Participants reported their current arousal and valence by choosing an item on the nine-point Likert scale. Section 2 began immediately. Participants saw a pair of eyes and several characteristics beneath it. Participants evaluated the stimulus based on the characteristics before proceeding to the next page. In Section 2, participants rated eight pairs of eyes regarding the perceived sex, age, attractiveness, trustworthiness, intelligence, realism, and familiarity. After the eighth stimulus was evaluated, participants entered Section 3. An instruction screen was presented to remind the participants the adjectives describing arousal and valence. Participants then evaluated the perceived arousal and valence on the same pairs of eyes they saw before, but in a different order. After evaluating the last stimulus, participants entered Section 4 to report their demographics. Section 1 was presented again to measure the post-survey arousal and valence. Finally, a thank you screen was shown to thank the participants for their contribution.

Some variables are now clarified to avoid confusion for the remaining manuscript. The SAM scale was used to measure participants’ arousal and valence (Section 1) and the perceived emotional state of the stimuli (Section 3). Participants’ arousal and valence were not analyzed in this manuscript since it was unrelated to the study goal. Therefore, arousal and valence refer to the participants’ ratings when participants evaluated the eyes.

### 2.5. Analysis

The analyses from both experiments were conducted in SPSS (IBM SPSS Statistics for Windows, Version 26.0). The dependent variables (DVs) and independent variables (IV) for both experiments were the same. The DVs were scores for each characteristic of sex, age, attractiveness, trustworthiness, intelligence, realism, familiarity, arousal, and valence. The IV was pupil size (large, medium, small) and vergence.

Data inclusion was run for Experiments 1 and 2 prior to other analyses. The goal was to analyze data from participants who did not fail the attention-check questions. In Experiment 1, data from 29 (10.94%) participants were not included since they selected the wrong items for the attention-check questions. Thus, we analyzed data of 236 (89.06%) participants for Experiment 1. In Experiment 2, data from three (2.73%) participants were rejected since they answered the attention-check question incorrectly. The remaining participants (*n* = 107, 97.27%) were included for the analysis.

Several variables were re-coded. The critical variable for comprehending this manuscript is the DV, sex. The Likert scale for sex ranged between very masculine to very feminine. This variable was re-coded for the analysis such that higher values indicate greater accuracy at identifying the sex of the eye models.

The goal of the study was to determine effect sizes of pupil size and vergence on the estimation of various characteristics. Both experiments were analyzed the same way following previous studies [26,31]. We evaluated the influence of the pupils on perceived characteristics by running a 4 × 1 repeated measures ANOVA for each DV. We conducted post-hoc pairwise comparisons for DVs with significant main effects to further investigate the specific contributions of the pupils on the ratings. A conservative approach to account for inflated α values was adopted. The significance value for the post-hoc tests was determined using Bonferroni-correction: 0.05 ÷ 6 = 0.008 across the four pupil sizes [32].

Statistical assumption violations were tested before running all analyses. Sphericity assumptions and appropriate corrections were used for the repeated measures ANOVA. We determined whether sphericity assumption was violated using Mauchly’s W [33]. Greenhouse-Geisser correction was applied when comparisons violated the sphericity assumption [34]. Results concerning the statistical assumption tests were reported in Appendix A to streamline the readability of the manuscript.

We ran an additional correlational analysis between the attractiveness, trustworthiness, and intelligence ratings since these ratings are often associated with each other [35,36]. We did not find any consistencies in correlations across these variables in both experiments. Results of the correlations for each experiment were reported in Appendix B.

## 3. Results

### 3.1. Pupils Do Not Influence Perceived Sex and Age

Figure 4a shows the ratings for perceived sex and age in Experiment 1. There were no differences for sex ratings, F(2.82, 662.01) = 0.31, *p* = 0.81, η_p_^2^ = 0.001, and age ratings, F(3, 705) = 1.21, *p* = 0.31, η_p_^2^ = 0.005. Figure 4b illustrates the perceived sex and age ratings in Experiment 2. We found no differences for sex ratings, F(2.74, 290.12) = 0.54, *p* = 0.66, η_p_^2^ = 0.005 and age ratings, F(3, 705) = 0.11, *p* = 0.95, η_p_^2^ = 0.001.

In summary, the pupils did not influence perceived sex and age ratings.

### 3.2. Unrealistic Pupils Influence Perceived Attractiveness, Trustworthiness, and Intelligence

Figure 5a visualizes the ratings of attractiveness, trustworthiness, and intelligence in Experiment 1. The main effect of attractiveness ratings reached significance, F(2.77, 650.38) = 36.85, *p* < 0.001, η_p_^2^ = 0.14. Post-hoc pairwise comparison was conducted for attractiveness ratings using Bonferroni-correction (*p* < 0.008) and the results were presented in Table 1. The comparisons showed that both big and medium pupils appeared more attractive than small pupils and vergence. Small pupils also looked more attractive than vergence ones.

There was a significant main effect for trustworthiness ratings, F(3, 705) = 40.87, *p* < 0.001, η_p_^2^ = 0.15. Post-hoc pairwise comparison was conducted with Bonferroni-correction (*p* < 0.008) and presented in Table 1. Both big and medium pupils looked more trustworthy than small pupils and vergence.

The main effect for intelligence ratings was statistically different, F(3, 705) = 32.57, *p* < 0.001, η_p_^2^ = 0.12. Post-hoc pairwise comparison using Bonferroni-correction (*p* < 0.008) was performed. The comparisons indicated that both big and medium pupils appeared more intelligent than small pupils and vergence. Small pupils also looked more intelligent than vergence.

Figure 5b shows the ratings for attractiveness, trustworthiness, and intelligence for Experiment 2. There were no differences for attractiveness ratings, F(2.31, 244.63) = 0.65, *p* = 0.58, η_p_^2^ = 0.006, and trustworthiness ratings, F(3, 705) = 0.91, *p* = 0.44, η_p_^2^ = 0.009.

There was a main effect for intelligence ratings, F(3, 705) = 3.20, *p* = 0.024, η_p_^2^ = 0.03. Post-hoc pairwise comparison was conducted using Bonferroni-correction (*p* < 0.008) and presented in Table 2. There were statistical differences in intelligence ratings between different pupil sizes. However, none of these comparisons survived Bonferroni-correction.

In summary, the pupils significantly influenced the ratings for attractiveness, trustworthiness, and intelligence for Experiment 1. Small and vergence pupils received the lowest ratings as compared to big and medium pupils. This result was not replicated in Experiment 2. Instead, there was only a significant main effect of perceived intelligence.

### 3.3. Unrealistic Pupils Influence Perceived Realism and Familiarity

Figure 6a illustrates the ratings for realism and familiarity in Experiment 1. Realism ratings were different, F(2.83, 665.13) = 99.27, *p* < 0.001, η_p_^2^ = 0.30. Post-hoc pairwise comparison using Bonferroni-correction was conducted (*p* < 0.008) and the results depicted in Table 3. The results illustrated that both big and medium pupils appeared more realistic than small pupils and vergence. Small pupils did not differ from vergence since the comparison did not survive Bonferroni-correction.

The main effect for familiarity ratings was significant, F(3, 705) = 30.62, *p* < 0.001, η_p_^2^ = 0.12. Post-hoc pairwise comparison using Bonferroni-correction (*p* < 0.008) was conducted and the results shown in Table 3. Again, both big and medium pupils looked more familiar than small pupils and vergence.

Figure 6b visualizes the realism and familiarity ratings for Experiment 2. The main effect for realism ratings was not significant, F(3, 705) = 2.18, *p* = 0.09, η_p_^2^ = 0.02. There was no significant main effect for familiarity ratings, F(3, 705) = 1.32, *p* = 0.27, η_p_^2^ = 0.01.In short, big and medium pupils appeared more realistic and familiar than small and vergence pupils for Experiment 1. This effect was absent in Experiment 2.

### 3.4. Pupils Do Not Influence Perceived Arousal, and Unrealistic Pupils Influence Perceived Valence

Figure 7a illustrates arousal and valence ratings in Experiment 1. There were no differences for arousal ratings, F(2.69, 631.62) = 1.65, *p* = 0.18, η_p_^2^ = 0.007. However, there were differences for valence ratings, F(2.90, 681.25) = 21.62, *p* < 0.001, η_p_^2^ = 0.08. Post-hoc pairwise comparison with Bonferroni-correction (*p* < 0.008) was run, and the results visualized in Table 4. The comparisons revealed that both big and medium pupils received greater valence ratings than small pupils and vergence.

Figure 7b pictures the arousal and valence ratings for Experiment 2. There was no significant main effect of arousal ratings, F(2.70, 286.03) = 0.78, *p* = 0.50, η_p_^2^ = 0.007. The main effect for valence ratings was not significant, F(2.74, 290.48) = 0.38, *p* = 0.38, η_p_^2^ = 0.009.

In conclusion, pupils did not influence arousal ratings in both experiments. However, the pupils influenced valence ratings in Experiment 1. Specifically, small and vergence pupils appeared more negatively in valence than big and medium pupils. This was not replicated in Experiment 2.

## 4. Discussion

Some studies show that a person’s characteristics, such as attractiveness or trustworthiness, could be perceived by looking at the eyes and the pupils [16,17]. In our study using only eyes with neutral expressions, we found large rating differences for exaggerated pupils (Experiment 1), but these differences were mostly absent for realistically looking pupils (Experiment 2). Pupils did not influence perceived sex and age, and they also did not systematically affect ratings of attractiveness, trustworthiness, intelligence, realism, familiarity, and valence. Contrary to the literature, we found no evidence of specific directional associations between pupils and ratings (i.e., big pupils denote higher ratings). Also, participants in both experiments could not perceive a person’s arousal from looking at the pupils. Therefore, it seems that the pupils, although when they are exaggerated could sometimes influence the perceived characteristics of another person, provide information mainly when interpreted in relation to other characteristics of facial expression.

### 4.1. How the Pupils Influence Perceived Characteristics of a Person

It seems unlikely that the pupils can influence the perceived sex and age of a person. In fact, we found no differences in sex and age ratings between the pupils. From the literature, biological sex influences eye size [6,37]. However, biological sex is unrelated to pupil size [38,39]. Age is typically perceived via features such as wrinkles [9,40]. However, pupil size is related to age. Older people have smaller pupils than younger people [41]. This decrease in pupil size is also very small [42] and difficult to perceive. We speculate that one could perceive age if the pupils appear cloudy (i.e., cataracts). This is due to the fact that cataracts are common in the older population [43]. Nonetheless, it is unlikely that sex and age can be perceived from changing pupil sizes.

Other characteristics about a person may be perceived by looking at the pupils, which could be driven by more complex processes. Our data show that the perceived attractiveness, trustworthiness, intelligence, realism, familiarity, and valence, can be influenced by pupil size when looking at neutral eyes. However, our results do not show that pupil sizes alone influence how a person reads various characteristics about another person by looking at the eyes.

Kret and De Dreu [17] showed that big (i.e., dilating) pupils appeared more attractive and trustworthy than small (i.e., constricting) pupils. In their experiment, big and small pupils differed by over 80% pupil area. If Kret and De Dreu’s results could be explained by pupil size differences, then we should also observe similar results between big and medium pupils in Experiment 1, since big and medium pupils differed by 100% in area. We found no differences between these two exaggerated pupils (Experiment 1). We also did not find any differences between big and small realistic pupils (Experiment 2) and they differed by twice the size (+100% area). Thus, it is unlikely that size alone can explain why the ratings differed from our study.

A key difference between our results and those from Kret and De Dreu’s study is motion. Although both studies used neutral eyes, our stimuli were static images, as compared to dynamic pupil changes. This hint that the changing of pupil sizes influences our perception regarding attractiveness and trustworthiness of another person. As a final remark, our results for exaggerated pupils changed in the same direction as those in Kret et al. [17], despite being static images. Hence, it is possible that there may be an interaction between static pupil sizes and motion from dynamically changing pupils, which contribute to how we read different characteristics about a person by looking at their eyes.

### 4.2. Ecological Validity of Image Manipulation in Experiments

The directional associations between pupils and ratings shown in previous studies [12,17,44] were missing from our study. Instead, in Experiment 1, eyes with small and vergence pupils were rated much lower than big and medium pupils in attractiveness, trustworthiness, intelligence, realism, familiarity, and valence. This finding was not replicated in Experiment 2 with more realistic pupils. This finding raises the question concerning to which extent ecological validity in stimulus preparation influences the generalizability of the results.

One explanation of the differences in our results might be related to the *uncanny valley* effect. This effect refers to evoked feelings of uneasiness when an artificial agent or imagery looks almost human-like but not perfectly [45]. It might be that in Experiment 1, the small pupils and the simulated vergence evoke *uncanny* vibes, lessening the trustworthiness of the eyes in comparison to big pupils. Importantly, big pupils also looked unrealistic, but was not susceptible to the uncanny valley effect. Participants could have rated big pupils as less unrealistic, but not considered them uncanny based on a general understanding of pupillary response. Big pupils are uncommon in well-lit conditions for most every-day situations. The pupils constrict and become comparably small in response to light or changes in illumination [46,47]. Therefore, the exaggerated big pupils, although appearing unrealistic in Experiment 1, were not prone to the uncanny valley effect.

Laboratory experiments often take place in highly controlled environments so that manipulations can be accounted for. However, lab environments hardly mimic the real-world. When interacting in the real world, we often integrate multisensory cues to minimize communication errors [48,49]. These multisensorial cues are often missing in experiments to simplify the number of variables needed to be manipulated. As a result, realistic stimuli often produce smaller effects compared to artificial/schematic ones [50]. In the context of our study, the eye-stimuli are also manipulated in unrealistic ways in existing literature, either by changing pupil sizes across an arbitrary duration [16,17], or by artificially modifying the sizes [12]. However, these manipulations are not replicable in the real world.

Pupillary changes are dynamic and unpredictable in the real world. The pupils can change very quickly in response to minor fluctuations in luminance, cognitive processes, and arousal [42,51,52]. Some studies suggest that our eyes can sense changes in another person’s pupil sizes through mimicking the pupil size of another person or animal during social interactions [16,53,54]. However, these studies cannot reliably isolate the influence of luminance on pupil dynamics [55,56]. In fact, pupil dynamics may not be easily seen in the real world since most (79%) people have brown irises compared to blue (10%) and other colors across the world [57]. There is also a spectrum of different shades for brown irises. Thus, it is difficult to say whether sensation of another person’s pupil size is possible despite the ever-changing luminance in the environment since there is a high chance that most people do not even see another person’s pupils due to the iris color.

It is unlikely that we can process changes in another person’s pupils unconsciously or consciously report the pupillary changes in real-world interactions. This is due to the fact that we do not make constant eye contact with whom we interact with. Eye contact can vary depending on the physical distance between people and the interaction’s phase. For example, eye contact is shorter and less frequent between the opposite sex, or when members are very close together [58]. In contrast, eye contact is more common when someone is listening to a speaker [59]. It is also unlikely that we pay close attention and look intently into another person’s eyes throughout the interaction as doing so causes the other person to feel uneasy. Since we do not make constant eye contact with others, it is less likely that we unconsciously process another person’s pupillary changes or become aware of such changes during real-world interactions. Hence, we can speculate that the impact of our measures may not replicate to real-world social interactions, since participants in our experiments were aware of the differently sized pupils and vergence when rating characteristics.

Lastly, an important aspect to consider is whether the stimulus preparation could be meaningful in drawing parallel conclusions about the real world. To that, some researchers have called for a clearer context-driven approach in defining ecological validity [60], where research questions should be defined as context-specific questions. In any case, future experiments should adopt realistic and multisensory stimuli, whilst formulating specific contexts, to further investigate the pupils’ role in influencing how we perceive characteristics about another person by looking at their eyes.

### 4.3. Perceiving Arousal from the Pupils

Pupil dilation indicates arousal when pupil size changes are tracked in an observer [61]. The pupils also dilate when we are emotionally stimulated [62]. It seems that we may not be good at perceiving a person’s emotional arousal when looking at their pupils. This is since we found no differences in arousal ratings for both experiments. One criticism to our claim could be since the exaggerated pupils appeared absurd in Experiment 1. However, we did not find any differences when using more realistic pupils in Experiment 2. The findings may change if we had presented emotionally expressive eyes. However, this would suggest that the perceived emotional arousal was extracted by integrating information about the eye expression and the pupil size. In short, it is likely that we are bad at perceiving a person’s emotional arousal from looking at their pupils.

Additional visual signals besides the pupils are required to perceive a person’s arousal, specifically the sexual arousal. The pupils dilate when a person is sexually aroused. The pupils dilate in response to sexual attraction [63]. Consequently, a person with larger pupils is perceived as more attractive [44,64]. One may even evaluate a stranger’s pupils in online dating to pick their potential partners [65]. These studies show that pupil sizes influence perceived arousal (in terms of sexual arousal) when a full face is visible. This differs from our study where only the eyes were shown. Since we did not measure perceived sexual arousal, it remains possible that sexual arousal could be perceived by looking at the pupils in the eyes.

### 4.4. The Pupils as Important Cues in Interactions

It is important to investigate how the pupils influence what we perceive about another person. The pupils tell us where the eyes are gazing [66,67], so they help us understand how the eyes inform mutual interaction [68]. In such interactions, pupil dilations communicate trust [69], eye blinks signal the transition of conversational topics [70], gaze patterns can indicate specific cognitive processes that are relevant for learning [71] and recognizing eye movement patterns facilitates the recognition of facial expressions [72]. Teachers who gaze into the students’ eyes during teaching are perceived to show greater interests in the student’s learning [73]. Gazing at students also helps capture the students’ attention [74] and fosters an interpersonal connection to their students, which is critical for the quality of teaching and learning [75,76]. With the recent surge in online-schooling, the frequency of making eye contact has fallen dramatically, creating severe problems in teaching as teachers must be able to observe their students’ attention to create an effective and efficient learning environment. In on-site teaching contexts, teachers can instantly pool information from many pupils to infer the direction of a group’s collective gaze [77]. In online learning, this is not possible (since the webcam is not embedded in the screen), so alternative measures need to be considered in online education, such as mapping the teachers gaze point on the material for guiding the students [78] or visualizing the aggregated gaze points of students [79].

### 4.5. Limitations and Future Directions

The current study investigates the pupils in neutral static eye images. Thus, the results are specific only to static eyes without expressions. It is known that we rely on the eyes to identify fear and surprise [80,81]. Dynamic changes in the eye region also influence whether one correctly recognizes certain emotional expressions [82]. Since the eyes also convey different emotions, it would be interesting to investigate how the interpretation of the current study changes, when eye emotions are introduced. Future studies could investigate the influence of varying pupils of either different static emotional eye expressions or dynamic eye expressions on the perceived characteristics of another person by looking at the eyes. The current sample sizes differ between the two experiments. Future experiments should aim for more balanced samples, especially when comparing data across experiments.

The addition of the first Purkinje image makes the eyes appear more lifelike. This is a technique commonly used in art. One criticism of Experiment 2 was that the reflections could have reduced the overall perceivable pupil size. According to the literature, the presence and absence of these reflections do not alter the perceived gaze of a painted portrait. It only affects the realism [83]. Therefore, subsequent studies can compare whether the first Purkinje images reduce the overall perceived pupil size.

Cultural factors influence how often a person looks at the eyes. Eastern cultures have smaller emphasis on the eyes than the western culture [84,85]. We did not query participants’ ethnicity. Participant recruitment was also of convenient means (i.e., social media, friends, families). We also did not test eyes from other ethnicities. Hence, our conclusions are limited to the western culture.

It is without a doubt challenging in capturing the pupil dynamics in a controlled and reliable manner so that they mimic real-world behaviors. Nevertheless, future studies should strive for more ecological valid stimuli, such as recording real pupil dynamics, and presenting them as stimuli. Studies should also define context-specific problems to solve, so that ecological validity could be captured more precisely. Alternatively, the study could be conducted on real-person interactions where the experimental conditions elicit dilated or constricted pupils.

## 5. Conclusions

The eyes are important in communication since they convey information about our cognitive and affective states. We investigated the effects of pupil size and vergence on how an observer reads various characteristics of another person by looking at the person’s neutral expression eyes. In exaggerated pupils, we found that small pupils and vergence were rated lower in attractiveness, trustworthiness, intelligence, realism, familiarity, and valence than large and medium pupils. These findings were absent for realistic pupils. The exaggeration could have induced some form of uncanniness, causing participants to rate them lower than realistic eyes. Regardless, our results indicate that the pupils may influence how we perceive a person’s characteristics, even when we look at emotionally neutral eyes. Therefore, the findings suggest that pupil size and vergence may be regarded as important social signals in subconscious social interaction processes. Importantly, future experiments need to pay stronger emphasis on ecologically valid stimuli when investigating the pupils and its relationship regarding the perceived characteristics of a person.

## Figures and Tables

**Figure 1 behavsci-12-00283-f001:**
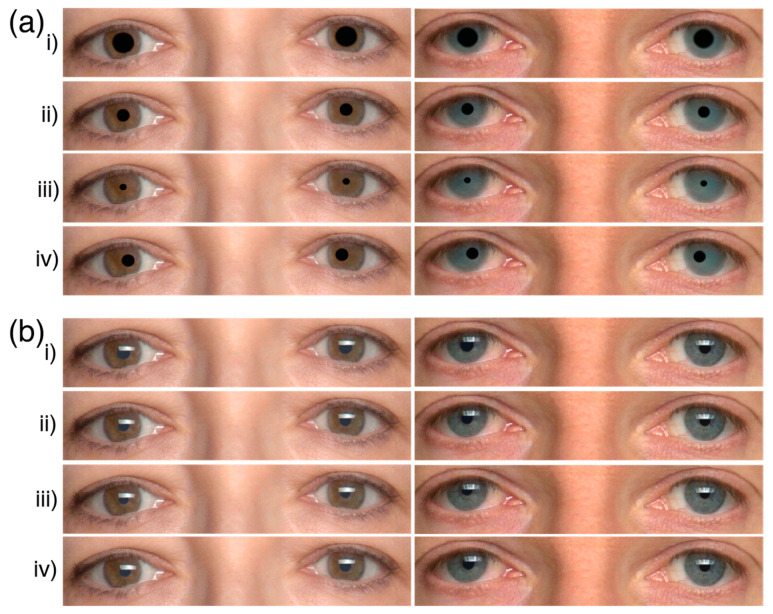
Illustrations of the stimuli used in (**a**) Experiment 1 and (**b**) Experiment 2. The eyes were 1140 × 226 pixels in dimensions. The pupils were (i) big, (ii) medium, (iii) small, and (iv) vergence. The pupils in (**a**) have the same area across all eye models. The pupils in (**b**) have different areas based on the eye models. The models in this illustration were models #29 and #70 from the FACES database.

**Figure 2 behavsci-12-00283-f002:**
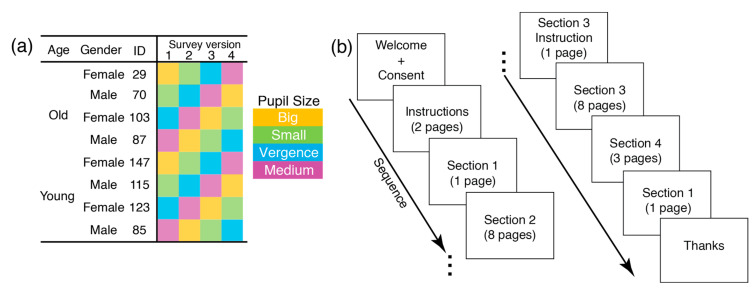
Survey version and sequence for both experiments. (**a**) illustrates the survey versions based on the Latin-square for presenting the stimuli. There were eight eye models. The age, gender, and ID of the models were counterbalanced. Each eye model never appeared more than once within a survey version. The colors correspond to various pupils. (**b**) shows the survey sequence. All participants completed the same sequence.

**Figure 3 behavsci-12-00283-f003:**
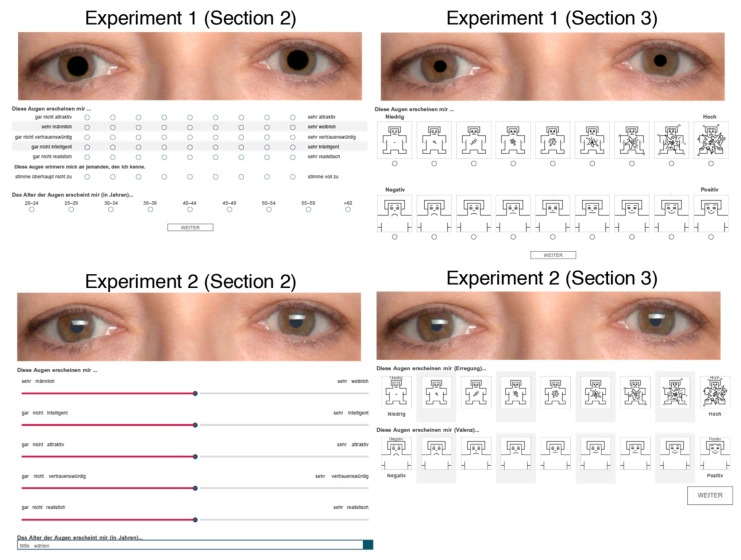
Sample screenshots of the survey layout in Sections 2 and 3 of the survey when the survey is not completed on a mobile device (i.e., mobile phone). In both experiments, the stimulus always appeared at the top of the screen. For illustration purposes, the eyes in Section 2 depict large pupils and those in Section 3 depict medium pupils.

**Figure 4 behavsci-12-00283-f004:**
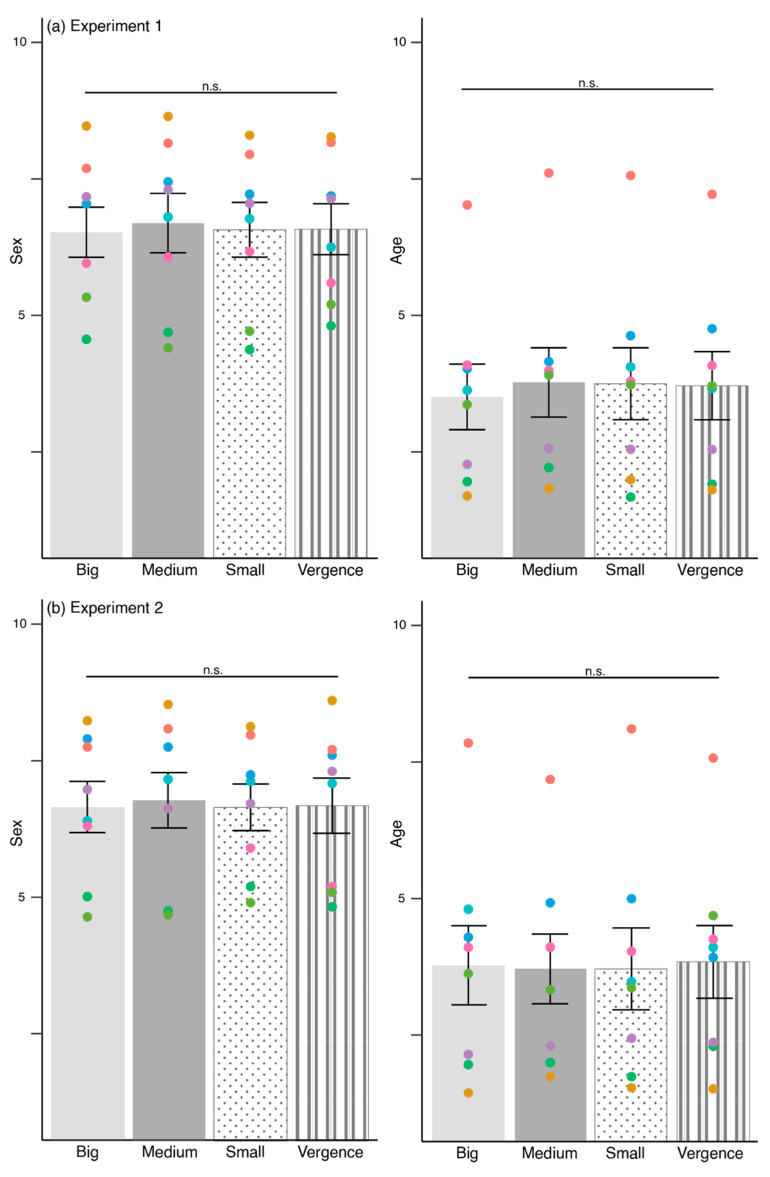
Sex and age ratings in (**a**) Experiment 1, and (**b**) Experiment 2 across big, medium, small, and vergence pupils. The colored scatter corresponds to each eye model. Error bars depict standard error of means, S.E.M. n.s.—non-significant.

**Figure 5 behavsci-12-00283-f005:**
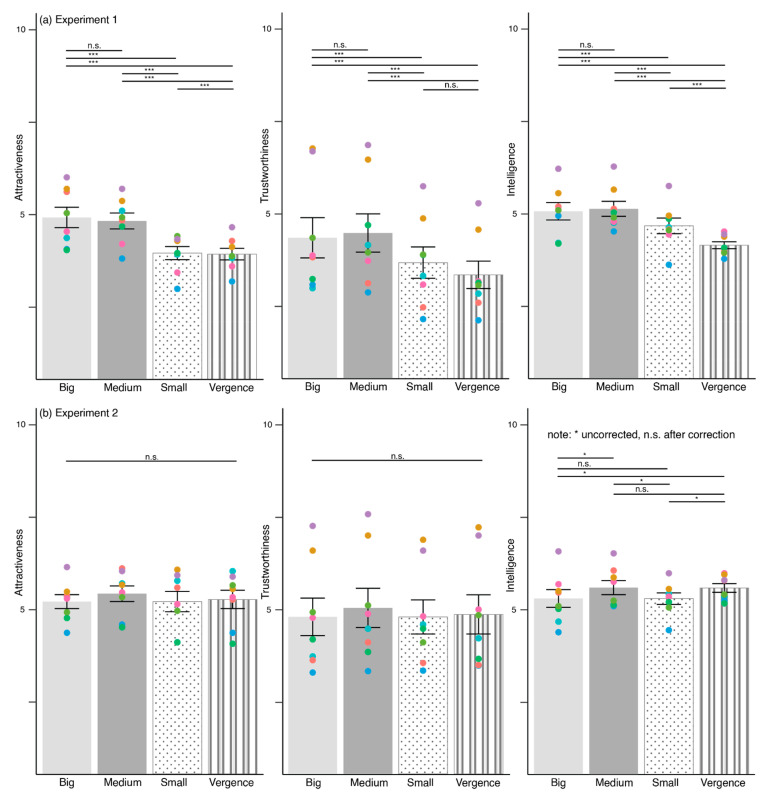
Attractiveness, trustworthiness, and intelligence ratings in (**a**) Experiment 1, and (**b**) Experiment 2 across big, medium, small, and vergence pupils. The colored scatter corresponds to each eye model. Error bars depict standard error of means, S.E.M. *** *p* < 0.001, * *p* < 0.05. Intelligence ratings in Experiment 2 are not significant after Bonferroni-correction (*p* < 0.008). n.s.—non-significant.

**Figure 6 behavsci-12-00283-f006:**
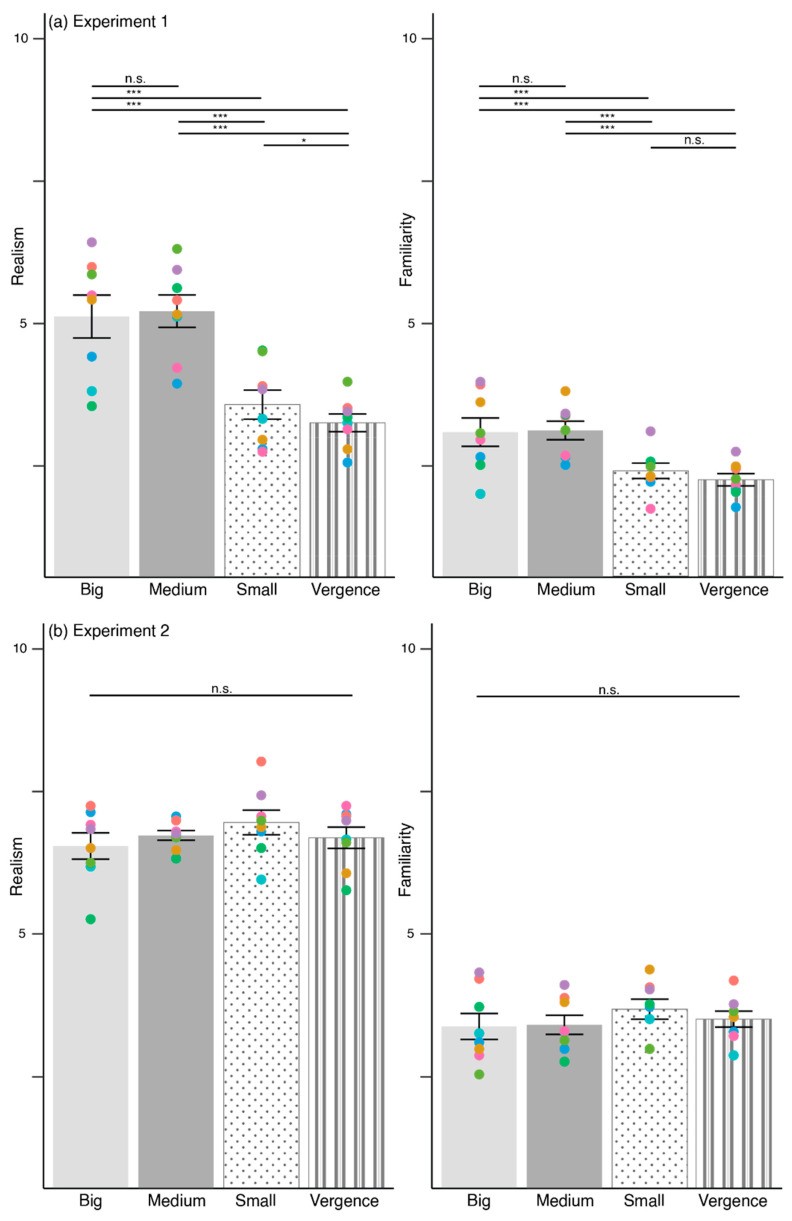
Realism and familiarity ratings in (**a**) Experiment 1, and (**b**) Experiment 2 across big, medium, small, and vergence pupils. The colored scatter corresponds to each eye model. Error bars depict standard error of means, S.E.M. *** *p* < 0.001, * *p* < 0.05, n.s.—non-significant.

**Figure 7 behavsci-12-00283-f007:**
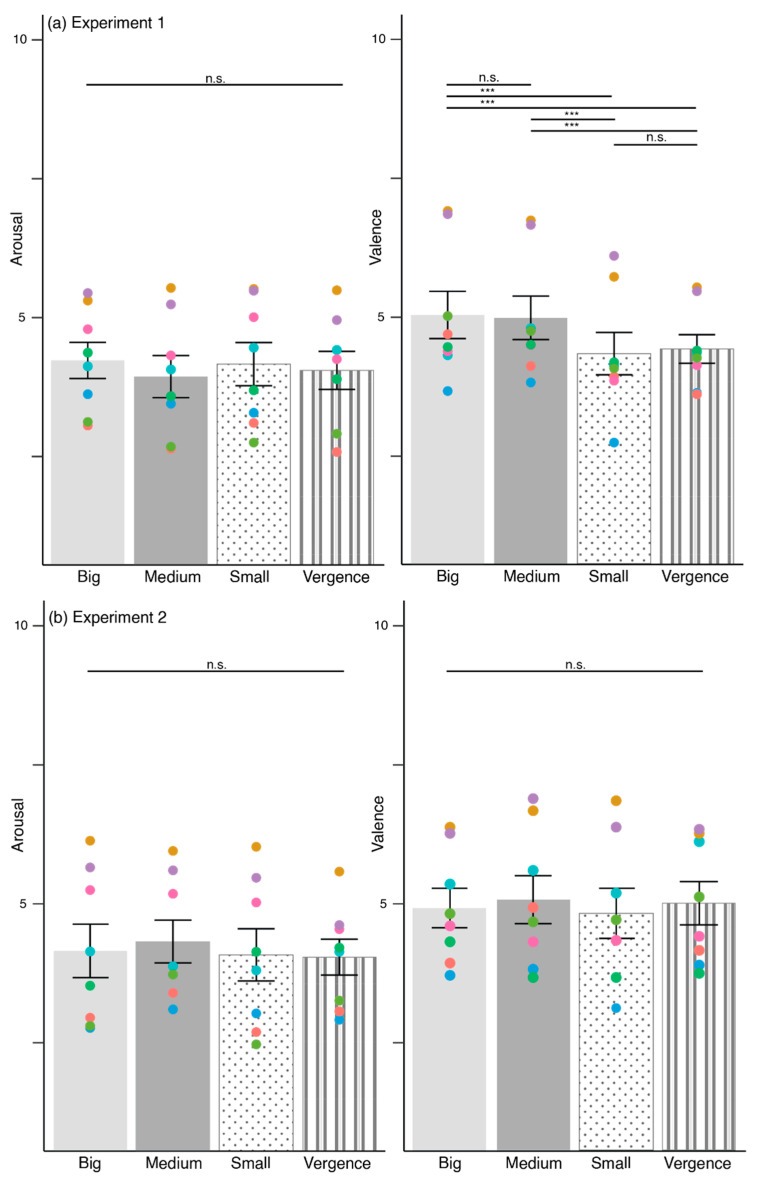
Arousal and valence ratings in (**a**) Experiment 1, and (**b**) Experiment 2 across big, medium, small, and vergence pupils. The colored scatter corresponds to each eye model. Error bars depict standard error of means, S.E.M. *** *p* < 0.001, n.s.—non-significant.

**Table 1 behavsci-12-00283-t001:** Post-hoc comparisons for attractiveness, trustworthiness, and intelligence ratings in Experiment 1.

Factor	Comparison Pair	*t*-Stat	*df*	*SD*	*p*	Cohen’s *d*
Attractiveness	Big (M = 4.34, *SD* = 1.68) & Medium (M = 4.50, *SD* = 1.51)	−1.09	235	2.20	0.28	0.07
	Big & Small (M = 3.71, *SD* = 1.56)	4.76	235	2.05	<0.001	0.31
	Big & Vergence (M = 3.33, *SD* = 1.63)	8.45	235	1.83	<0.001	0.55
	Medium & Small	7.12	235	1.71	<0.001	0.46
	Medium & Vergence	9.09	235	1.97	<0.001	0.59
	Small & Vergence	3.00	235	1.90	0.003	0.20
Trustworthiness	Big (M = 4.93, *SD* = 1.63) & Medium (M = 4.83, *SD* = 1.40)	0.80	235	1.82	0.42	0.05
	Big & Small (M = 3.95, *SD* = 1.67)	7.33	235	2.05	<0.001	0.48
	Big & Vergence (M = 3.93, *SD* = 1.56)	8.36	235	1.83	<0.001	0.54
	Medium & Small	7.19	235	1.88	<0.001	0.47
	Medium & Vergence	8.15	235	1.70	<0.001	0.53
	Small & Vergence	0.17	235	1.78	0.87	0.01
Intelligence	Big (M = 5.08, *SD* = 1.40) & Medium (M = 5.16, *SD* = 1.30)	−0.72	235	1.76	0.47	0.05
	Big & Small (M = 4.69, *SD* = 1.34)	3.60	235	1.63	<0.001	0.23
	Big & Vergence (M = 4.16, *SD* = 1.60)	7.79	235	1.81	<0.001	0.51
	Medium & Small	4.26	235	1.68	<0.001	0.28
	Medium & Vergence	8.69	235	1.77	<0.001	0.57
	Small & Vergence	4.65	235	1.77	<0.001	0.30

**Table 2 behavsci-12-00283-t002:** Post-hoc comparisons for intelligence ratings in Experiment 2.

Factor	Comparison Pair	*t*-Stat	*df*	*SD*	*p*	Cohen’s *d*
Intelligence	Big (M = 5.32, *SD* = 1.00) & Medium (M = 5.61, *SD* = 1.17)	−2.12	235	1.39	0.036	0.21
	Big & Small (M = 5.32, *SD* = 1.03)	0.04	235	1.21	0.97	0.004
	Big & Vergence (M = 5.60, *SD* = 1.17)	−2.29	235	1.27	0.024	0.22
	Medium & Small	−2.22	235	1.35	0.028	0.22
	Medium & Vergence	0.03	235	1.55	0.98	0.003
	Small & Vergence	−2.29	235	1.29	0.024	0.22

**Table 3 behavsci-12-00283-t003:** Post-hoc comparisons for realism and familiarity ratings in Experiment 1.

Factor	Comparison Pair	*t*-Stat	*df*	*SD*	*p*	Cohen’s *d*
Realism	Big (M = 5.13, *SD* = 2.17) & Medium (M = 5.22, *SD* = 1.90)	−0.58	235	2.23	0.56	0.04
	Big & Small (M = 3.56, *SD* = 2.11)	10.14	235	2.37	<0.001	0.66
	Big & Vergence (M = 3.26, *SD* = 2.11)	11.42	235	2.51	<0.001	0.74
	Medium & Small	13.35	235	1.90	<0.001	0.87
	Medium & Vergence	13.67	235	2.19	<0.001	0.89
	Small & Vergence	2.15	235	2.13	0.03	0.14
Familiarity	Big (M = 3.08, *SD* = 1.89) & Medium (M = 3.12, *SD* = 1.84)	−0.30	235	1.87	0.77	0.02
	Big & Small (M = 2.40, *SD* = 1.47)	5.57	235	1.86	<0.001	0.36
	Big & Vergence (M = 2.24, *SD* = 1.61)	7.17	235	1.80	<0.001	0.47
	Medium & Small	6.47	235	1.69	<0.001	0.42
	Medium & Vergence	7.58	235	1.77	<0.001	0.49
	Small & Vergence	1.50	235	1.67	0.13	0.10

**Table 4 behavsci-12-00283-t004:** Post-hoc comparisons for valence ratings in Experiment 1.

Factor	Comparison Pair	*t*-Stat	*df*	*SD*	*p*	Cohen’s *d*
Valence	Big (M = 5.03, *SD* = 1.32) & Medium (M = 5.00, *SD* = 1.25)	0.27	235	1.67	0.79	0.02
	Big & Small (M = 4.35, *SD* = 1.42)	5.81	235	1.79	<0.001	0.38
	Big & Vergence (M = 4.41, *SD* = 1.37)	5.55	235	1.70	<0.001	0.36
	Medium & Small	6.03	235	1.65	<0.001	0.39
	Medium & Vergence	4.91	235	1.83	<0.001	0.32
	Small & Vergence	−0.59	235	1.59	0.55	0.04

## Data Availability

The data and analyses scripts associate with this manuscript are available from: Lau, Wee Kiat (2022): The role of realism in perceiving neutral expression eyes. Open Access Repositorium der Universität Ulm und Technischen Hochschule Ulm. http://dx.doi.org/10.18725/OPARU-42603 (accessed on 22 June 2022).

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
