# Peer review of "Small Pupils Lead to Lower Judgements of a Person’s Characteristics for Exaggerated, but Not for Realistic Pupils"

_behavsci, 2022, doi:10.3390/bs12080283_

Round 1

Author Response

Dear anonymous reviewer 1, we thank you for your comments. They were helpful to improve the manuscript. Please see the attachment. Comments are colored black, replies are colored blue, and manuscript revisions are colored green.  

Reviewer 2 Report

Summary

The authors investigated the effects of pupil size and vergence on how an observer reads other peoples’ characteristics in neutral expression of the eyes. The tested characteristics were sex, age, attractiveness, trustworthiness, intelligence, realism, familiarity, valence, and arousal. In artificial looking pupils (i.e., when black disks were substituting pupils, Experiment 1), small pupils and vergence were related to lower attractiveness, trustworthiness, intelligence, realism, familiarity, and valence than large and medium pupils. However, these findings were not replicated with realistic pupils in Experiment 2. This study comprised of two methodologically sound online experiments with good sample sizes, but its contributions to the existing literature remain unclear. I suggest the authors should strengthen the motivation of the study and the justification of the measured characteristics. Findings related to the ecological validity of the stimuli (i.e., comparison between Experiment 1 and 2) are interesting, but I’m not certain about the scientific relevance of the manipulation used in Experiment 1. I also have some concerns related to analyses. I’ve listed my comments according to the sections and hope these would be helpful for the authors.

General comments

Abstract

The abstract should describe the manipulations and results in more detail.

Introduction

In general, the introduction would benefit from a better layout between prior work and the present study. Previous findings are listed in the introduction, but it remains unclear what gaps in the literature the current study aims to fill in. I also think some of the references may not be needed, e.g., it’s not clear how the gaze-cueing effects are relevant here.

p 3. Lines: 99-100: What was the reason for including these nine factors to the study?

Methods

How was the sample size determined?

The use of mobile phone was allowed in responding to the survey. Was there a way to separate responses based on the device type used for giving the responses and did it affect the results?

What was the purpose of showing unrealistic pupils in Experiment 1?

Analysis: Were the counterbalancing versions (Figure 2a) included in the analysis as group factors to check whether the versions differed in terms of ratings?

Results

Were the factors correlated with each other? Based on Figure 5 and the ANOVA results, it seems that attractiveness and trustworthiness are highly correlated, and that intelligence also shows a similar pattern. To my knowledge these concepts are related to each other, i.e., attractive people are more easily perceived as intelligent and trustworthy. I think this aspect should be addressed.

The results are hard to follow. Instead of repeating the same sentence structure for different factors the significant results should be highlighted. It would be easier if for example the post-hoc statistics were shown in a table instead of the main text.

Discussion

p. 15, lines 464-466: I can’t follow the logic of this reasoning: “it is important to note that we did not replicate the results of the existing literature: Kret and De Dreu [21] showed that big (i.e., dilating) pupils appeared more attractive and trustworthy than small (i.e., constricting) pupils.” On lines 468-469 the authors state that in their results “small pupils appeared less attractive and trustworthy than big and medium pupils (Experiment 1)”. Isn’t this essentially the same effect but only for artificial looking pupils.

Specific comments

p 1. lines 30-31: some words may be missing here.

Results: To simplify the text, it is not necessary to repeat the word significant many times since the reader can judge this based on the p-value.

Author Response

Dear anonymous reviewer 2, we thank you for your comments. They were helpful to improve the manuscript. Please see the attachment. Comments are colored black, replies are colored blue, and manuscript revisions are colored green.  

Round 2

Reviewer 2 Report

The authors have responded adequately to most of my comments.

However, I still think that the Abstract lacks details of pupil manipulations in Experiment 2. I also suggest that the power calculations and correlations between rating results should be added to the manuscript. I think these would be helpful for the readers.

Author Response

Dear anonymous reviewer 2, please see the attachment for the formatted responses. Once again, we thank you for your comments in helping to improve the manuscript. Please note that comments are colored black, the responses are colored blue, and the revisions are colored green.

This manuscript is a resubmission of an earlier submission. The following is a list of the peer review reports and author responses from that submission.

Round 1

Reviewer 1 Report

Comments to authors:

In this manuscript the authors present a study to assess whether the different size of pupil’s dilatation could influence the perception of different aspect of a person (sex, age, attractiveness, trustworthiness, intelligence, realism, familiarity, arousal, and valence). In the experiment 1, the authors used stimuli with an exaggerate manipulation of the pupil’s size, while in the experiment 2 they used a more realistic manipulation.

In Exp1 small and vergence pupils received lower ratings in attractiveness, trustworthiness, intelligence, familiarity and realism compared to medium and big pupils. While no effects were found in Exp2. In addition, the pupils from Exp1 were lower rating in attractiveness, trustworthiness, intelligence, familiarity, realism and valence than the Exp2.

The manuscript is well written, the design and the analyses are overall appropriate to address the experimental question.

I have one major revision and some minor observations, mainly aimed at improving the quality of the manuscript before publication.

Participant.

Most participants that took part in the study were females, do the authors think that this aspect may in some affect the result? Have the author think about using the sex of the subject as a covariate?

Material.

The major concern about the study, is the type of material used here. I have to say I am not an expert in the field. However, it seems to me, that the stimuli used in the Exp1 were way too unrealistic, then I assume that it could be pretty obvious that eyes so unreal would be perceived as less attractive, trustworthy, intelligent, etc.

I don’t think that authors can infer that pupil’s dilatation influence the perception of the characteristic of a person, because they found differences only with unrealistic stimuli, but not with more realistic eyes, which the size of the pupil was also manipulated.

The authors say, regarding the stimuli in Exp2 that “The camera glare was included to preserve the realism of 175 the original photos (see Figure 1b)”, and it is fine, however, I would have included the camera glare even in the stimuli of Exp1, to reduce, at least, their unrealism. In this way, I think that the stimuli from the two experiment could have been more comparable, still maintaining their substantial differences.   

Also, without the camera glare in the stimuli in Exp1, I don’t think that authors can conclude that the difference in the 2 experiments is due to the realism in the pupils rather than the realism of the eyes pictures in general.

Figure

In the file that I received, I couldn’t properly see the figure 5, since it was displaced to the right. Then, almost half part of the intelligent bar graph didn’t appear. Despite of this problem, I think that the * and the n.s. used by authors, can be misleading to understand the figure. I suggest to report only the significant results, through the *, after the Bonferroni correction is applied, and to mention it in the caption of all the figure.

Discussion

The results evidenced that the stimuli in the Exp1 are perceived as less familiar than the stimuli in the Exp2. It is likely that stimuli so unrealistic would be judged as less familiar, still, I think the authors should add a brief statement to emphasize this aspect.